# Simulating the interplay of dipolar and quadrupolar interactions in NMR by spin dynamic mean-field theory

Timo Gräßer[1,*], Götz S. Uhrig[1],

**1** Condensed Matter Theory, TU Dortmund University, Otto-Hahn Straße 4, 44221 Dortmund, Germany
* timo.graesser@tu-dortmund.de

August 25, 2025

## ABSTRACT

The simulation of nuclear magnetic resonance (NMR) experiments is a notoriously difficult task, if many spins participate in the dynamics. The recently established dynamic mean-field theory for high-temperature spin systems (spinDMFT) represents an efficient yet accurate method to deal with this scenario. SpinDMFT reduces a complex lattice system to a time-dependent single-site problem, which can be solved numerically with small computational effort. Since the approach retains local quantum degrees of freedom, a quadrupolar term can be exactly incorporated. This allows us to study the interplay of dipolar and quadrupolar interactions for any parameter range, i.e., without the need for a perturbative treatment. We highlight the relevance of local quantum effects by a comparison with the classical analogue system.

## I. INTRODUCTION

Atomic nuclei with a spin $S > 1/2$ experience an electric quadrupolar interaction in an anisotropic electronic environment. This is relevant in the broad field of nuclear magnetic resonance (NMR) [1], but also in the spin dynamics in charged quantum dots [2–7]. Discovered in the 1930s, NMR has continuously evolved into a widely used technique in material physics, chemistry, biology and medicine. The principle of NMR is to resonantly address the nuclear spins of a sample in order to gain information about the sample's composition and molecular structure. This is based on the fact that nuclear spins interact with their chemical environment through the associated magnetic moments. Important contributions to the nuclear spin dynamics include the chemical shift, the J-coupling, the dipole-dipole interaction and the aforementioned quadrupolar interaction [8]. The latter two are the focus of this article. The dipolar interaction couples nearby nuclear spins to one another proportional to $1/r^3$, where $r$ is the relative distance. This entails a many-body problem. The quadrupolar interaction, on the other hand, is completely local. It results from a coupling between the electric quadrupole moment of a deformed atomic nucleus with an electric field gradient generated by the surrounding electron cloud [1].

In general, the simulation of a large spin system represents a notoriously difficult task. As the Hilbert space grows exponentially with the system size, exact simulations [9] are only feasible for a few tens of spins and therefore suffer from finite-size effects. In many systems, dipolar interactions are well captured by classical simulations [10, 11]. In this case, the computational effort grows only polynomially with the system size so that, in practice, finite-size effects can be essentially removed. Despite this clear advantage, it is not *a priori* clear how well the classical approximation works in a specific geometry. The accuracy is expected to be reduced in low-dimensional systems or systems with well-separated, small groups of spins, where quantum effects tend to be more relevant. Hybrid quantum-classical approaches can assist to some degree [12], but as they do not make use of translational invariance, they can become quite demanding. Besides this, it is not clear, how well an additional quadrupolar interaction can be treated in classical or hybrid simulations. The key question is how important the local quantum nature of the spins is.

In many scenarios, the quadrupolar interaction strongly dominates the dipolar one [8], so that the effect of the latter can essentially be neglected. But there exist also systems, where the interplay of the quadrupolar and dipolar interaction is relevant. A prominent example is ⁷Li [13, 14] where homonuclear dipolar interactions have been observed to significantly affect stimulated-echo spectra [15, 16]. Often, a perturbative treatment of the dipolar interaction [17, 18] or an exact simulation of a few adjacent spins [16, 19] suffices to capture the main physics. However, it is not clear how reliable such treatments are if the quadrupolar and dipolar interaction are of the same order.

In this article, we introduce spin dynamic mean-field theory, short spinDMFT, as an alternative approach for simulating the interplay of dipolar and quadrupolar interactions. SpinDMFT is developed for dense spin systems at infinite temperature [20]. "Dense" in this context means that the approach is accurate in the limit where each spin has an infinite number of interaction partners. "Infinite temperature" corresponds to the thermal energy being much

larger than any internal energy scale of the considered system. Then, the initial statistical operator corresponds to completely disordered spins. On the one hand, this is a strong constraint, but on the other hand it makes spinDMFT perfectly tailored to the field of NMR because nuclear spins are disordered in most experiments due to the smallness of their gyromagnetic ratios. A strong advantage of spinDMFT is that it requires only small computational effort, which allows for systematic extensions such as cluster spinDMFT [21] and non-local spinDMFT [22]. Moreover, the method is highly versatile because it works with an effective single-site Hamiltonian. This easily allows for the inclusion of local spin terms such as local magnetic fields, static or time dependent, as well as quadrupolar interactions.

The article is set up as follows. First, we formulate a basic model for a spin system containing a dipolar and quadrupolar interaction in Sec. II. Subsequently, in Sec. III A, we apply spinDMFT to this model obtaining a single-site model that can be solved numerically. The results are presented and discussed in Secs. III B and III C. In Sec. IV, we draw a comparison to the classical analogue system. Finally, the article is concluded in Sec. V.

## II. BASIC MODEL

We consider a high-temperature nuclear spin ensemble of homogeneous, spatially-fixed spins with $S > 1/2$. The ensemble shall be subject to a strong magnetic field as usual in NMR experiments. The spins interact with one another via the secular homonuclear Hamiltonian [8]

$$\mathbf{H}_{\text{DD}} = \frac{1}{2} \sum_{i,j} d_{ij} \left( 2\mathbf{S}_i^z \mathbf{S}_j^z - \mathbf{S}_i^x \mathbf{S}_j^x - \mathbf{S}_i^y \mathbf{S}_j^y \right). \tag{1}$$

with

$$d_{ij} := d_{\vec{r}_{ij}}(\vec{n}_B) = \frac{1 - 3\left(\vec{n}_{ij} \cdot \vec{n}_B\right)^2}{2} \frac{\mu_0}{4\pi} \frac{\gamma_i \gamma_j \hbar^2}{|\vec{r}_{ij}|^3}, \qquad \vec{n}_{ij} := \frac{\vec{r}_{ij}}{|\vec{r}_{ij}|}, \qquad \vec{n}_B := \frac{\vec{B}}{|\vec{B}|}, \tag{2}$$

where $\vec{r}_{ij} = \vec{r}_j - \vec{r}_i$ is the distance vector between spins $i$ and $j$ and $\vec{B}$ is the magnetic field. Any self-interactions are ruled out, i.e., we set $d_{ii} := 0$. In Eq. (1) and henceforth, we label operators by boldface symbols. Each nucleus locally interacts with an electric field gradient (EFG), which is captured by the secular quadrupolar interaction term [8]

$$\mathbf{H}_{\text{Q}} = \Omega \sum_i \left( 3\mathbf{S}_i^{z\,2} - \vec{\mathbf{S}}_i^2 \right) \tag{3}$$

with

$$\Omega := \frac{3eQ}{4S(2S-1)} V^{zz}(\vec{n}_B), \tag{4}$$

where $Q$ is the electric quadrupole moment of the nucleus and $V^{zz}$ is the $zz$-component of the EFG tensor. The square of the spin vector operator is a constant and thus corresponds to a constant energy shift, which is irrelevant for the spin dynamics and will be omitted henceforth. The Hamiltonian

$$\mathbf{H} = \mathbf{H}_{\text{DD}} + \mathbf{H}_{\text{Q}} \tag{5}$$

describes a complex many-body quantum system, which cannot be exactly solved for large numbers of spins. An approximation has to be made.

## III. SPIN DYNAMIC MEAN-FIELD THEORY

### A. Single-site model and closed self consistency

Spin dynamic mean-field theory is an elegant and efficient way to describe this many-body system approximately [20]. The approximation is reliable if each spin has a large number of interaction partners. This implies that the fields describing the local spin environments (henceforth called local-environment fields) defined by

$$\vec{\mathbf{V}}_i(t) := \sum_j d_{ij} \underline{\underline{D}} \, \vec{\mathbf{S}}_j, \qquad \underline{\underline{D}} = \begin{pmatrix} -1 & 0 & 0 \\ 0 & -1 & 0 \\ 0 & 0 & 2 \end{pmatrix}, \tag{6}$$

consist of many contributions. A measure for the number of contributions is the effective coordination number [20]

$$z_{\text{eff}} := \frac{\left(\sum_j d_{ij}^2\right)^2}{\sum_j d_{ij}^4}, \tag{7}$$

which is independent of $i$ as the system is homogeneous. If $z_{\text{eff}}$ is large ($\gtrsim 5$), it is well-justified to replace each local-environment field by a dynamic Gaussian mean-field $\vec{V}(t)$. This results in a local mean-field Hamiltonian

$$\mathbf{H}^{\text{mf}}(t) = \vec{V}(t) \cdot \vec{\mathbf{S}} + 3\Omega \mathbf{S}^{z\,2}. \tag{8}$$

The mean-field is zero on average and its second moments result from the self-consistency condition [20]

$$\overline{V^\alpha(t)V^\beta(0)}^{\text{mf}} = J_{\text{Q}}^2 \delta^{\alpha\beta} D^{\alpha\alpha\,2} \langle \mathbf{S}^\alpha(t)\mathbf{S}^\alpha(0)\rangle, \tag{9}$$

where $D^{\alpha\alpha}$ are diagonal matrix elements of $\underline{\underline{D}}$ and we defined the quadratic coupling constant

$$J_{\text{Q}}^2 := \sum_j d_{ij}^2. \tag{10}$$

The formal equation to compute the spin autocorrelations is given by

$$\langle \mathbf{S}^\alpha(t)\mathbf{S}^\alpha(0)\rangle = \int \mathrm{D}\mathcal{V}\, p(\mathcal{V})\, \langle \mathbf{S}^\alpha(t)\mathbf{S}^\alpha(0)\rangle^{\text{loc}}(\mathcal{V}). \tag{11}$$

Here, $\mathcal{V}$ is a mean-field time series and $p(\mathcal{V})$ its multivariate Gaussian probability distribution. The expectation value $\langle \mathbf{S}^\alpha(t)\mathbf{S}^\alpha(0)\rangle^{\text{loc}}(\mathcal{V})$ is carried out in the Hilbert space of a single spin considering the time evolution generated by the Hamiltonian in Eq. (8) for a specific $\mathcal{V}$. Since the system is at infinite temperature, the density matrix is proportional to the identity in all considered expectation values.

The defined self-consistency problem is solved by numerical iteration. Starting from an initial guess for the spin autocorrelations, one computes the second mean-field moments via Eq. (9) and uses them to update the spin autocorrelations by means of Eq. (11). This process is repeated until the spin autocorrelations are converged, which requires only about 5 iteration steps. In practice, the path integral in Eq. (11) is evaluated by discretizing the time and applying a Monte-Carlo simulation. For more details on the derivation and numerical implementation of the approach, we refer to the original article in Ref. [20].

### B. Results in the time domain

Figure 1 displays the converged results of the normalized autocorrelations

$$G^{\alpha\alpha}(t) := \frac{3}{S(S+1)} g^{\alpha\alpha}(t), \tag{12}$$

with

$$g^{\alpha\alpha}(t) := \langle \mathbf{S}^\alpha(t)\mathbf{S}^\alpha(0)\rangle \tag{13}$$

for different spin lengths $S$ and quadrupolar interaction strengths in the time domain. The top panels each show the transverse autocorrelation $G^{xx} = G^{yy}$ and the bottom panels the longitudinal autocorrelation $G^{zz}$. Any off-diagonal autocorrelations vanish due to rotational symmetry about the $z$-axis. The time is given in units of $1/\tilde{J}_{\text{Q}}$ defining

$$\tilde{J}_{\text{Q}} := \sqrt{\frac{S(S+1)}{3}}\, J_{\text{Q}}, \tag{14}$$

where $J_{\text{Q}}$ is the quadratic coupling constant defined in Eq. (10) and $\hbar$ is set to one. We choose this specific spin-dependent rescaling for better comparison of the results of different spin lengths. Increasing $S$ means increasing the strength of the mean-field and thus the speed of the decay. This scaling effect is compensated when depicting the time in units of $1/\tilde{J}_{\text{Q}}$.

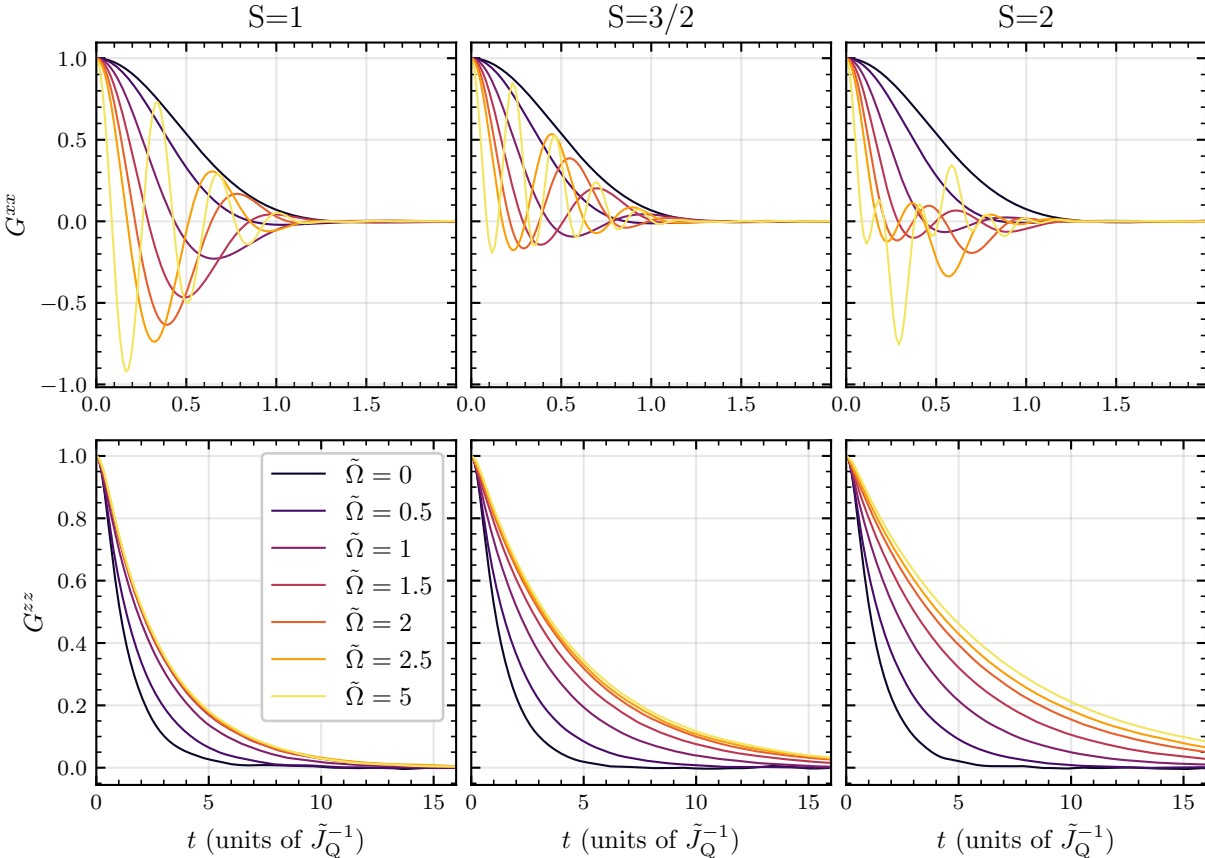

FIG. 1. Results for the normalized spin autocorrelation $G^{\alpha\alpha}$ from spinDMFT for different spin lengths and quadrupolar interaction strengths in the time domain. The top row shows the transverse and the bottom row the longitudinal results. The spin length is increased from left to right. Different quadrupolar strengths are indicated by different colors according to the provided legend.

Yet, the results for different spin lengths are visibly different. The quadrupolar interaction leads to oscillations in the transverse autocorrelations. This behavior is similar to a Larmor precession due to a magnetic field in $z$-direction. However, it is clearly not the same as the quadrupolar interaction term is quadratic in $\mathbf{S}^z$ in contrast to the Zeeman term. The oscillation frequencies depend on the quadrupolar interaction strength

$$\tilde{\Omega} := \frac{\Omega}{J_{\mathrm{Q}}}. \tag{15}$$

In the case of $S = 2$, two oscillations with different frequencies are overlapping. This behavior is best understood in the spectra, which will be considered in the next subsection. The longitudinal results do not oscillate, but nevertheless depend on the quadrupolar interaction strength. They show a monotonic decay which slows down upon increasing $\tilde{\Omega}$. This is not surprising as the quadrupolar interaction destabilizes the transverse spin components so that they average out faster. In return, the longitudinal correlation decays slower because its decay is driven by the transversal components. A qualitatively similar behavior was obtained when adding a static Gaussian noise in the $z$-direction, see Ref. [20].

For large values of $\tilde{\Omega}$, the longitudinal correlations depend only weakly on $\tilde{\Omega}$. We analyze this further by fitting exponentials $\exp(-t/T)$ to the tails of the results. The fits work exceptionally well, see Fig. 2 and Tab. I. The extracted relaxation times barely change when $\tilde{\Omega}$ is large, but they do not reach saturation as can be seen in Fig. 3. In the limit $\tilde{\Omega} \to \infty$, the mean-field becomes arbitrarily weak in comparison to the quadrupolar interaction which leads to a constant longitudinal autocorrelation. The convergence to this limit case is very slow as is demonstrated by Fig. 3.

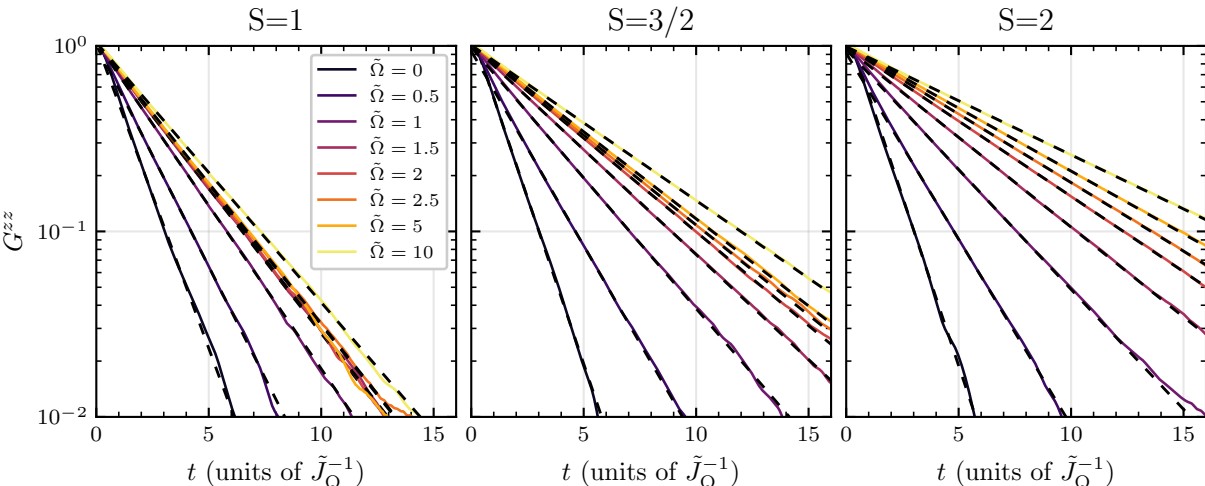

FIG. 2. Results of the longitudinal spin autocorrelations in the time domain in logarithmic representation. The dashed lines display exponential fits, which work exceptionally well. The random oscillations of the data result from the statistical error of the Monte-Carlo simulation.

TABLE I. Extracted decay times of the longitudinal autocorrelations $G^{zz}$ in dependence of $\tilde{\Omega}$ for different spin lengths. The corresponding exponential fits are shown in Fig. 2. The numerical errors are in the order of $10^{-2} - 10^{-3}$.

| $\tilde{\Omega}$ | $S = 1$ $T/\tilde{J}_{\mathrm{Q}}^{-1}$ | $S = 3/2$ $T/\tilde{J}_{\mathrm{Q}}^{-1}$ | $S = 2$ $T/\tilde{J}_{\mathrm{Q}}^{-1}$ |
|---|---|---|---|
| 0 | 1.32 | 1.21 | 1.19 |
| 0.5 | 1.81 | 2.07 | 2.16 |
| 1 | 2.47 | 3.08 | 3.36 |
| 1.5 | 2.78 | 3.84 | 4.48 |
| 2 | 2.79 | 4.30 | 5.35 |
| 2.5 | 2.86 | 4.47 | 5.88 |
| 5 | 2.87 | 4.62 | 6.41 |
| 10 | 3.10 | 5.18 | 7.44 |

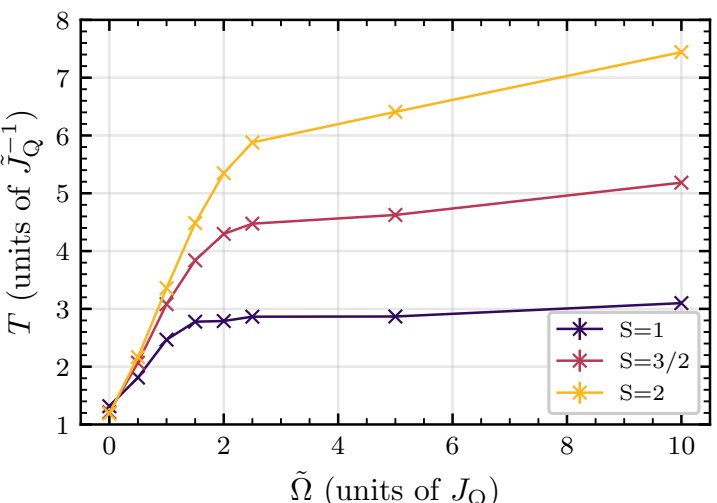

FIG. 3. Plot of the extracted decay times of the longitudinal autocorrelations $G^{zz}$ in dependence of $\tilde{\Omega}$ for different spin lengths. The corresponding exponential fits are shown in Fig. 2.

## C. Results in the frequency domain

The spectra are shown in Fig. 4 for different spin lengths and strengths of the quadrupolar coupling. They are obtained by fast Fourier transform of the symmetrized temporal results. We stress that they are not referring to the free-induction decay (FID), but to the spin autocorrelation. The latter, however, can be considered a first-order approximation of the FID. The FID includes pair correlations in addition. These are not directly accessible in single-site spinDMFT, but may be computed by the extension non-local spinDMFT [22]. This is beyond the scope of the present article.

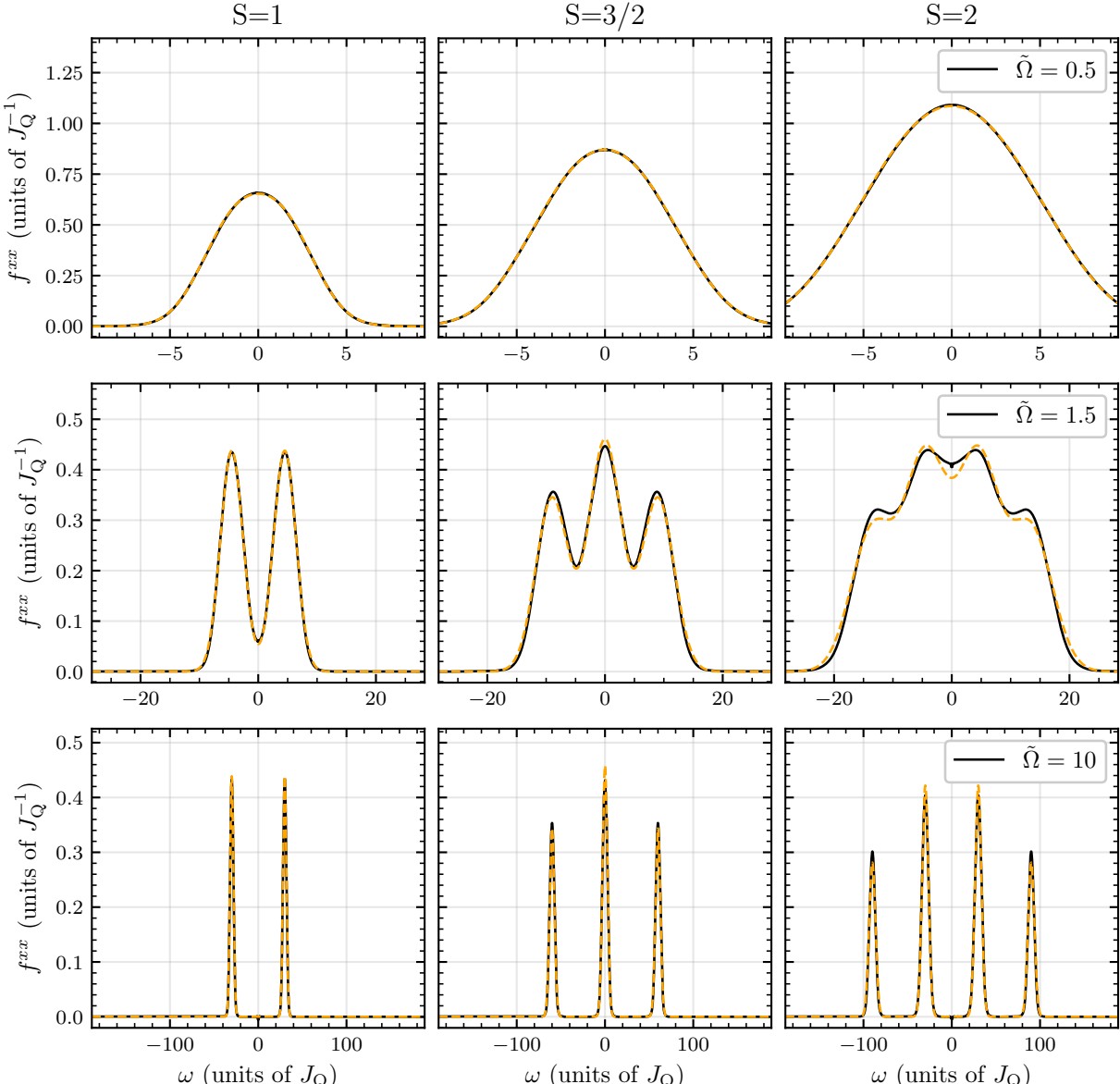

FIG. 4. Fourier transform $f^{xx}(\omega) \coloneqq \int_{-\infty}^{\infty} e^{-i\omega t} g^{xx}(t) dt$ of the transverse spin autocorrelation $g^{xx}(t)$ for different quadrupolar interaction strengths and spin lengths. The spin length is increased from left to right and the quadrupolar interaction from top to bottom. The orange dashed line corresponds to the Gaussian fit described in Eq. (17). Small deviations are seen at some of the peak maxima. These become smaller when allowing for an individual amplitude $A_i$ for each peak in the fit function. However, we prefer the shown fits because they require only a single parameter, namely, the standard deviation $\sigma$.

The autocorrelation spectra already provide very valuable information. The peaks are located rather precisely at

$$\omega_i = 6\Omega\left(S - i + \frac{1}{2}\right), \qquad\qquad i \in \{1, \ldots, 2S\}. \tag{16}$$

This can be understood from the local quantum mechanics: The peaks correspond to the expected quadrupolar transitions, see Fig. 5 for visualization. The advantage of spinDMFT consists in the *ab initio* prediction of the continuous line shapes induced by the dipolar interactions.

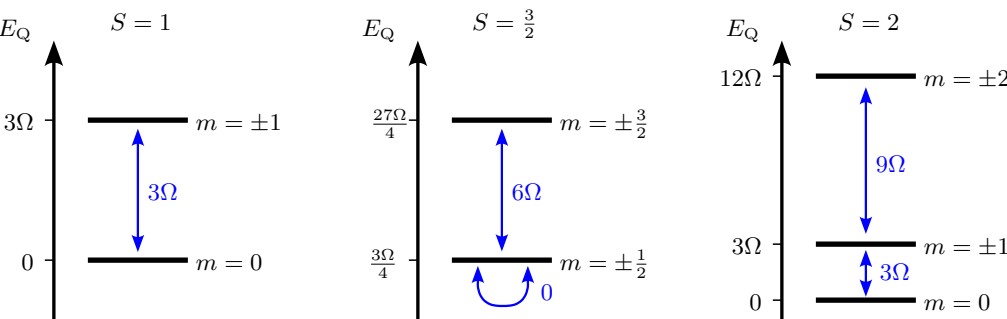

FIG. 5. Possible quadrupolar transitions with $\Delta m = \pm 1$ for different spin lengths. The quadrupolar energy is given by $E_{\text{Q}} = 3\Omega m^2$.

Remarkably, we find that spinDMFT predicts a Gaussian broadening of the resonance lines over the full considered parameter range. This can be well seen from the dashed lines in Fig. 4 which correspond to the fit function

$$f_\sigma(\omega) = \pi \sum_{i=1}^{2S} \frac{i}{2}\left(1 - \frac{i}{2S+1}\right) \frac{\exp\left(-\frac{(\omega - 3\Omega(2S - 2i + 1))^2}{2\sigma^2}\right) + \exp\left(-\frac{(\omega + 3\Omega(2S - 2i + 1))^2}{2\sigma^2}\right)}{\sqrt{2\pi\sigma^2}}, \tag{17}$$

which is derived from considering the exact local result and replacing the $\delta$-functions by Gaussian distributions. The fit parameters are listed in Tab. II. To account for the main effect of the spin length, the standard deviations are provided in units of $\tilde{J}_{\text{Q}}$. In this unit, they depend only weakly on the spin length $S$ and the quadrupolar interaction strength $\tilde{\Omega}$.

TABLE II. Resulting fit parameters for the spectra shown in Fig. 4 corresponding to the fit function in Eq. (17). The numerical errors are in the order of $10^{-3} - 10^{-4}$.

| $\tilde{\Omega}$ | $S = 1$ $\sigma/\tilde{J}_{\text{Q}}$ | $S = 3/2$ $\sigma/\tilde{J}_{\text{Q}}$ | $S = 2$ $\sigma/\tilde{J}_{\text{Q}}$ |
|---|---|---|---|
| 0.5 | 2.228 | 2.212 | 2.208 |
| 1.5 | 2.337 | 2.448 | 2.553 |
| 10 | 2.332 | 2.447 | 2.521 |

The longitudinal autocorrelations are well captured by exponential fits in the time domain, see Fig. 2. Therefore, their spectra are very well described by Lorenz curves $\Gamma/(\Gamma^2 + \omega^2)$ with decay rates $\Gamma = 1/T$. We refrain from showing the corresponding plots. The decay times $T$ are shown in Tab. I.

## IV.   COMPARISON TO CLASSICAL DYNAMICS

As pointed out in Ref. [20], the dynamics of a single spin in a classical mean-field is essentially classical. This is because the spin's equation of motion is linear in spin operators which makes it equivalent to that of a classical spin according to Ehrenfest's theorem. However, with the bilinear quadrupolar term included, this conclusion does not hold anymore. Despite being completely local, spinDMFT captures beyond-classical behavior for finite quadrupolar interactions.

To highlight the importance of simulating the local degrees of freedom quantum mechanically, we perform a simulation of the classical analogue system for comparison. The classical equation of motion can be derived as in Ref. [23]

$$\frac{\partial \vec{S}}{\partial t} = \frac{\partial H}{\partial \vec{S}} \times \vec{S}. \tag{18}$$

Inserting the mean-field Hamiltonian from Eq. (8), we obtain

$$\frac{\partial \vec{S}}{\partial t} = \vec{V}(t) \times \vec{S} + 6\Omega S^z \begin{pmatrix} -S^y \\ S^x \\ 0 \end{pmatrix},\tag{19}$$

which is equivalent to the quantum equation of motion, when replacing $\vec{S} \to \mathbf{S}$ and symmetrizing the last term. To simulate the classical dynamics, we average over the mean-field at all times as well as over the initial spin values at $t = 0$. The mean-field average works in the same way as for the quantum case. For the spin average, we fix the spins length to $\sqrt{S(S+1)}$ ensuring

$$\overline{S^{\alpha 2}} = \langle \mathbf{S}^{\alpha 2} \rangle\tag{20}$$

and choose the initial orientation uniformly distributed over the Bloch sphere due to the high-temperature limit.

The difference between quantum and classical dynamics already becomes apparent when considering the effect of a varied spin length $S$. The obtained equation of motion can be rewritten as

$$\frac{\partial \vec{n}}{\partial \tilde{t}} = \vec{W}(\tilde{t}) \times \vec{n} + 6\Omega n^z \begin{pmatrix} -n^y \\ n^x \\ 0 \end{pmatrix}\tag{21}$$

using the renormalized quantities

$$\vec{n} := \frac{1}{\sqrt{S(S+1)}}\vec{S}, \qquad \vec{W} := \frac{1}{\sqrt{S(S+1)}}\vec{V}, \qquad \tilde{t} := \sqrt{S(S+1)}t.\tag{22}$$

Note that a renormalization of $\vec{S}$ automatically implies a renormalization of $\vec{V}$ due to the self-consistency condition. The consequence of Eq. (21) is that the dynamics is independent of the spin length except for a rescaling of the time axis. This fact is already in stark contrast to the quantum results shown in the previous section, where changing the spin length leads to qualitatively different behavior beyond a sheer scaling factor.

The universal classical results for arbitrary spin length are presented in Fig. 6 versus the time. Both the transverse and longitudinal autocorrelations clearly differ from the quantum results shown in Fig. 1. The difference to the quantum mechanical results is the smallest for the largest value of the spin, $S = 2$. This is not surprising because larger spins tend to behave more and more classically.

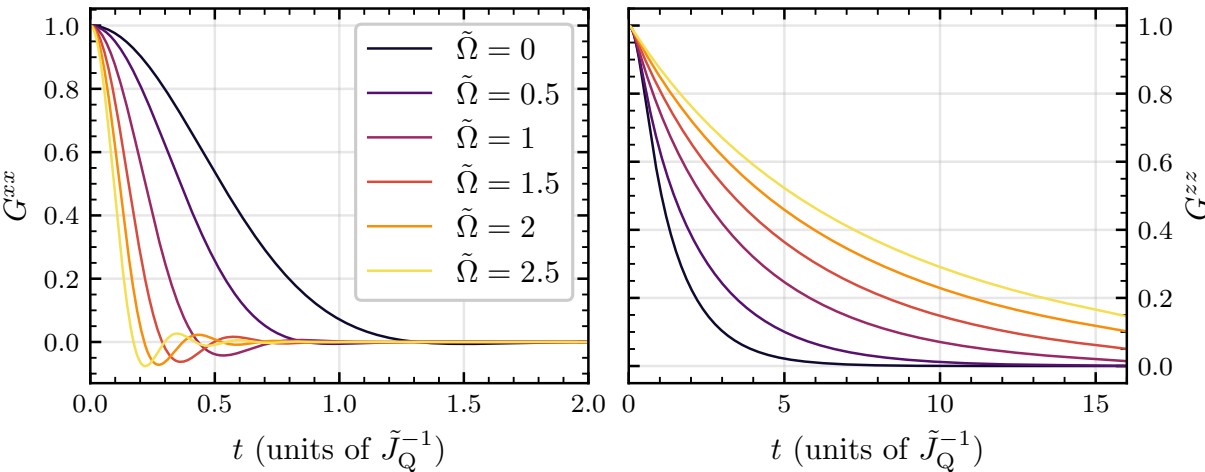

FIG. 6. Universal results of the spin autocorrelations from classical spinDMFT presented in the main text. The spin length is incorporated in the time axis which is given in inverse units of the rescaled quadratic coupling constant $\tilde{J}_{\mathrm{Q}}$.

This can be underlined analytically by the Frobenius norm

$$\|\mathbf{A}\| := \frac{1}{d}\mathrm{Tr}\{\mathbf{A}^{\dagger}\mathbf{A}\},\tag{23}$$

of an operator $\mathbf{A}$, where $d$ denotes the Hilbert space dimension. Similar to the consideration in Ref. [24], we compare the norm of the commutator of two spin components

$$\left\|\left[\mathbf{S}^\alpha, \mathbf{S}^\beta\right]\right\| = \|\mathbf{S}^\alpha\| = \frac{S(S+1)}{3}, \qquad\qquad \alpha \neq \beta, \tag{24}$$

to the norm of a product of two spin components

$$\left\|\mathbf{S}^\alpha \mathbf{S}^\beta\right\| = \frac{1}{15} S(S+1)\left(S(S+1) + \frac{1}{2}\right), \qquad\qquad \alpha \neq \beta. \tag{25}$$

We find that the relative commutator norm is suppressed by

$$\frac{\left\|\left[\mathbf{S}^\alpha, \mathbf{S}^\beta\right]\right\|}{\left\|\mathbf{S}^\alpha \mathbf{S}^\beta\right\|} \propto \frac{1}{S^2}. \tag{26}$$

Hence, the error from commuting non-commuting spin operators becomes smaller and smaller if $S$ is increased resembling more and more classical behavior. As can be seen in Fig. 7, this behavior is confirmed numerically by a direct comparison of the quantum results for increasing spin length with the classical ones.

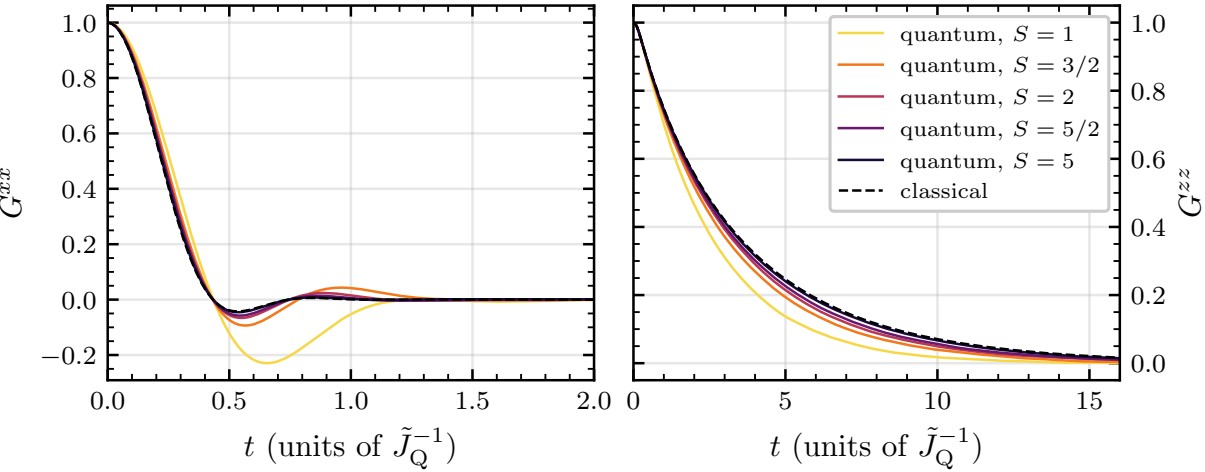

FIG. 7. Comparison of quantum spinDMFT for different spin lengths with the universal result of classical spinDMFT for a fixed quadrupolar coupling strength $\tilde{\Omega} = 1$.

In Fig. 8, we provide the classical spectra of the transverse autocorrelation. As the classical spin is continuous, there is an infinite number of possible transitions so that even for large $\Omega$, no peak structure is obtained in contrast to the quantum case. Instead, one obtains a superposition of an infinite number of peaks. For $\Omega \ll \sigma$, these peaks are very close to one another which leads to shapes strongly resembling Gaussian curves in total, since the individual peaks are Gaussian. For $\Omega \gg \sigma$, the shape of an individual peak becomes unimportant and only its position and weight matters. Starting from the exact local result for quantum spins (see Eq. (17) with the Gaussian functions replaced by $\delta$-distributions), and considering the limit of large $S$ in the step from Eq. (27a) to Eq. (27b), it can be shown that

$$
\begin{aligned}
F_{\text{class}}^{xx}(\omega) &= \frac{3}{S(S+1)}\pi \sum_{i=1}^{2S} \frac{i}{2}\left(1 - \frac{i}{2S+1}\right) \\
&\qquad \times \left[\delta\big(\omega - 3\Omega(2S - 2i + 1)\big) + \delta\big(\omega + 3\Omega(2S - 2i + 1)\big)\right] \tag{27a} \\
&= \frac{\pi}{16S(S+1)} \int_{1-2S}^{2S-1} \mathrm{d}x(2S + 1 - x)\left(1 + \frac{x}{2S+1}\right) \\
&\qquad \times \left[\delta\left(\frac{\omega}{3\Omega} - x\right) + \delta\left(\frac{\omega}{3\Omega} + x\right)\right] \tag{27b} \\
&= \frac{1}{\tilde{J}_Q \tilde{\Omega}} \frac{\pi}{4\sqrt{3}}\left[1 - \left(\frac{\omega/\tilde{J}_Q}{6\sqrt{3}\tilde{\Omega}}\right)^2\right] \Theta\left(6\sqrt{3}\tilde{\Omega} - \omega/\tilde{J}_Q\right) \Theta\left(6\sqrt{3}\tilde{\Omega} + \omega/\tilde{J}_Q\right). \tag{27c}
\end{aligned}
$$

This analytical result is shown in Fig. 8 by the orange dashed line. In case of $\Omega \gg \sigma$, the classical spectrum consists of a single parabola instead of distinct Gaussian peaks. This further underlines the need for a quantum-mechanical simulation of quadrupolar systems. A calculation treating all spins as classical vectors can capture the spin dynamics only for large spins at best.

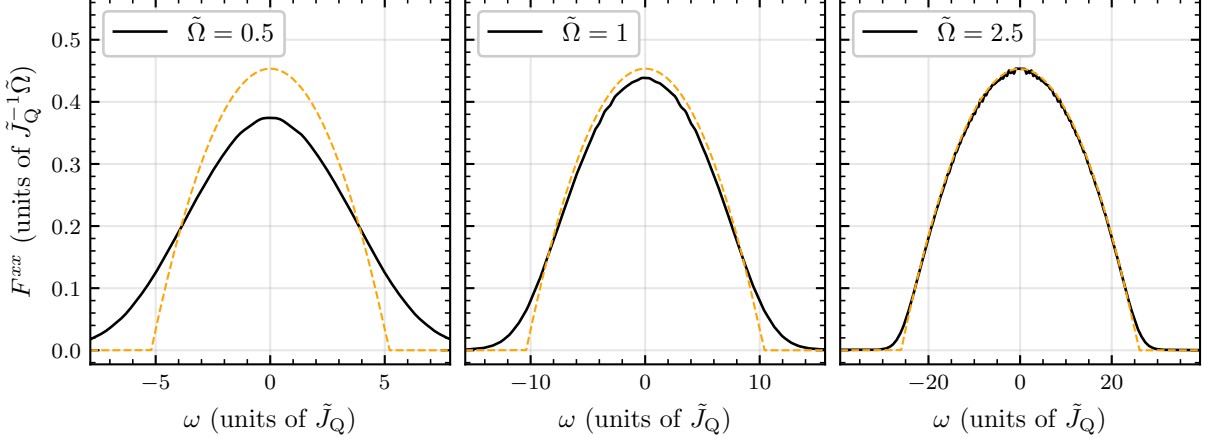

FIG. 8. Fourier transform $F^{xx}(\omega) \coloneqq \int_{-\infty}^{\infty} \mathrm{e}^{-\mathrm{i}\omega t} G^{xx}(t)\mathrm{d}t$ of the classical transverse spin autocorrelation $G^{xx}(t)$ for different quadrupolar interaction strengths. The spin length is incorporated in the units through $\tilde{J}_{\mathrm{Q}}$. The signal $F^{xx}$ is shown in units of $\tilde{J}_{\mathrm{Q}}\tilde{\Omega}$ to enhance the comparability of the results.

## V. CONCLUSION

Spin dynamic mean-field theory (spinDMFT) is an efficient numerical approach to compute the dynamics of dipolar spin systems at high temperatures. The key idea is to replace the environment of a spin by a time-dependent mean-field which is Gaussian distributed. This allows one to define a single-site problem and a self-consistency condition, which connects the variances of the mean-field to the spin autocorrelations. In this article, we showed how a quadrupolar interaction can be incorporated in spinDMFT: Since the quadrupolar term is completely local, it can be directly and exactly added to the single-site model.

The numerical evaluation yields an exponential longitudinal relaxation and an oscillating transverse relaxation. The latter is best understood in the frequency spectrum, where peaks can be identified at the expected quadrupolar transitions with $\Delta m = \pm 1$. According to spinDMFT, the dipolar interaction broadens the resonance lines to Gaussian functions over the full range of parameters, i.e., spin lengths and quadrupolar interaction strengths. Remarkably, a fit with a single parameter, namely, the peak standard deviation, suffices for an adequate description of the spectrum.

Another goal of this article was to draw a comparison to the classical analogue system. While classical simulations usually capture purely dipolar systems well, it turns out that the presence of a quadrupolar interaction precludes a classical description. The local spin degrees of freedom need to be simulated quantum-mechanically.

Typically, dipolar interactions are considered deleterious in NMR experiments due to the induced line broadening and associated difficulties in obtaining information from the spectra. However, if the dipolar line broadening can be predicted, systems with moderate dipolar interactions become accessible. As we demonstrated in this article, spinDMFT can be a suitable prediction tool for this scenario. We emphasize that the computational effort of spinDMFT is small allowing for extensive parameter sweeps and/or various extensions to increase the accuracy of the approach and access more complex systems. The mean-field framework can be extended to include inhomogeneous systems with several spin species as well as explicit time dependencies, such as pulses. We also highlight non-local spinDMFT [22] for computing FID's because spinDMFT itself can only access spin autocorrelations, which cannot directly be measured in experiment.

In future works, spinDMFT could be extended to magic-angle spinning (MAS) in order to study residual dipolar broadening [25, 26]. This would also allow comparison to and prediction of several experiments measuring MAS spectra of quadrupolar nuclei [14, 27]. Since quadrupolar interactions and explicit time-dependencies are accessible, spinDMFT could also be a useful simulation tool to study motion in Li-ion conductors [28, 29]. The results presented pave the way to a quantitative analysis and understanding of NMR results in a large variety of experiments.

## ACKNOWLEDGMENTS/FUNDING INFORMATION

We are thankful to R. Böhmer for useful discussions. Furthermore, we acknowledge funding by the Deutsche Forschungsgemeinschaft (DFG) in project UH90/14–2.

## CODE AVAILABILITY

A code collection concerning spinDMFT and its extensions has been published under `https://doi.org/10.17877/TUDODATA-2025-MD4EYWOL`.

---

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
