# Peer review of "Simulating the interplay of dipolar and quadrupolar interactions in NMR by spin dynamic mean-field theory"

_SciPost Physics_

## Round 1 · Referee Report · Anonymous (Referee 1) · 2025-9-19

Strengths

1) The manuscript is well written and structured, mostly self-contained and pedagogical. It should hence be accessible even for non-specialists. 2) The comparison of the results obtained from quantum and classical treatment of the spins allows to appreciate the importance of the dynamical treatment brought about by spinDMFT.

Weaknesses

1) Even though the limitations of the technique due to its inherent approximations are clearly stated, it is less clear how well the technique performs for calculating spin autocorrelations as relevant to NMR measurements. Since this is the motivation of the present manuscript, an exemplary model system with dipole-dipole and quadrupolar interaction that can be directly discussed in comparison to existing NMR measurements would be helpful in this context.

Report

The manuscript discusses an extension of the authors’ recently developed spinDMFT technique to include local quadrupolar interaction terms in the spin Hamiltonian. It is well written, rather pedagogical and accessible for non-specialists. Moreover, the comparison to classical limiting cases is convincing and I have no doubt on the validity of the presented results and the study itself. The present manuscript discusses a timely question in the context of NMR measurements and has clear potential for follow-up work. I therefore think that one of the acceptance criteria of SciPost Physics is clearly met.
However, even though the work is motivated by NMR, a direct (even qualitative) comparison to experimental data is unfortunately absent. This is why I would not go as far as qualifying it as a breakthrough on a long-standing research stumbling block as listed in the author indications on fulfilling the journal expectations.
All in all, I recommend the manuscript for publication in SciPost Physics after minor revisions, see requested changes below.

Requested changes

1) Statistical errors of the Monte Carlo (MC) simulation are mentioned explicitly in Fig. 2, but they should lead to error bars for most of the shown quantities. The authors might want to either include error bars in the respective figures or state clearly that those error bars would be smaller than the symbol size of the data points. 2) In Sec. III.B the authors fit exponentials to the longitudinal correlations, stating in Table I that numerical errors are in the order of $10^{-2}-10^{-3}$. They might want to clarify whether these errors refer to the quality of the fits or whether they also include the MC error. 3) Why is the spinDMFT broadening Gaussian and not e.g. Lorentzian? Is it rooted in the use of a Gaussian probability functional? What is the interpretation of this ‘Gaussian prediction’ of the line shape and has it been observed experimentally via NMR? The authors might want to comment on these questions in the manuscript. 4) The steps in between Eqns. (27a-c) should be explained in more detail.

Minor details: 5) Eqn. (22) lacks two right brackets ")". 6) In Fig. 8, the authors should mention the dashed orange line (analytic result of eqn. (27)) in the caption. 7) I would also like to ask the authors to check again the prefactor of Eqn. (27b). I may be wrong, but I think it lacks a factor $2/\vert \omega\vert$.

Recommendation

Ask for minor revision

  • validity: high
  • significance: high
  • originality: high
  • clarity: high
  • formatting: excellent
  • grammar: excellent

Author:  Timo Gräßer  on 2025-10-30  [id 5969]

(in reply to Report 1 on 2025-09-19)

General reply:

First of all, we thank the Referee for the very thorough reading of our manuscript and the constructive comments. We are very glad that the paper is perceived as "well written" and recommended for publication in SciPost Physics.

Furthermore, we fully understand the Referee's concern regarding the lack of a comparison to experiment. However, we would like to emphasize that the purpose of this work is a proof of principle. Our main goal was to demonstrate that quadrupolar interactions can be directly included in spinDMFT without changing the approach itself. Of course, a comparison to experiment is desirable, but unfortunately, we did not find a suitable and simple benchmark system with available experimental data in the literature. Measurements on quadrupolar spins in monocrystals are rarely performed and nowadays NMR experiments are rather complex (involving for example magic-angle spinning), which requires one to include many further details in the simulations complicating a direct comparison.

1. requested change:

Statistical errors of the Monte Carlo (MC) simulation are mentioned explicitly in Fig. 2, but they should lead to error bars for most of the shown quantities. The authors might want to either include error bars in the respective figures or state clearly that those error bars would be smaller than the symbol size of the data points.

Reply:

We fully agree that numerical errors should be discussed more thoroughly. The Monte-Carlo errors were already smaller than any linewidths except for the logarithmic plot. However, the error from finite time discretization, which we did not yet consider in detail, turned out to be relevant for large Omega. We performed additional simulations to obtain error estimates for this. Unexpectedly, we obtained changes in Fig. 2 and 3 and Tab. 1 for large Omega, when making the time discretization finer. The new plots suggest a saturation of the decay time for large Omega, which has not been discussed in the literature before. We assign this saturation to an effective dynamics that can be accessed by average Hamiltonian theory in the limit of large quadrupolar interactions.

Action taken:

We included error estimates or error bars in the respective figures and tables. We added a short paragraph about the effective dynamics in the limit of large quadrupolar interactions.

2. requested change:

In Sec. III.B the authors fit exponentials to the longitudinal correlations, stating in Table I that numerical errors are in the order of 10−2−10−3. They might want to clarify whether these errors refer to the quality of the fits or whether they also include the MC error.

Reply:

The errors referred to the quality of the fits. Indeed, we changed the fitting procedure in Fig. 2 from a two-parameter to a one-parameter fit which turned out to be significantly more robust at the price of yielding a slightly worse agreement for small values of Omega.

Action taken:

We changed the fitting procedure as described above. We decided to include estimates of the numerical error, which is dominated by the time discretization, instead of estimates of the fit quality.

3. requested change:

Why is the spinDMFT broadening Gaussian and not e.g. Lorentzian? Is it rooted in the use of a Gaussian probability functional? What is the interpretation of this ‘Gaussian prediction’ of the line shape and has it been observed experimentally via NMR? The authors might want to comment on these questions in the manuscript.

Reply:

We thank the referee for pointing this out. The Gaussian probability functional results from the central limit theorem for the mean-fields and it does not directly imply that the line shapes are Gaussian. Yet, the separation of time scales of transverse and longitudinal spin dynamics matters: it is induced here by the anisotropy of the dipolar interaction as well as by the purely longitudinal quadrupolar term. Thus, the longitudinal mean-fields are dominant and have a very long correlation time, i.e., they are rather static. This makes the dynamics comparable to that of an Ising model, where the line shapes are exactly Gaussian if there are many interaction partners so that the central limit theorem applies. We found a reference measuring NMR spectra in aluminium nitride, where indeed Gaussian dipolar line broadening is seen. However, we stress that the scenario is not equivalent to our model as it involves two spin types and autocorrelation and FID spectra are distinct from each other.

Action taken:

We added a paragraph about this in the manuscript.

4. requested change:

The steps in between Eqns. (27a-c) should be explained in more detail.

Reply:

We agree that the derivation was too brief.

Action taken:

We added further explanation sentences and an intermediate calculational step.

5. requested change:

Minor details: 5) Eqn. (22) lacks two right brackets ")". 6) In Fig. 8, the authors should mention the dashed orange line (analytic result of eqn. (27)) in the caption. 7) I would also like to ask the authors to check again the prefactor of Eqn. (27b). I may be wrong, but I think it lacks a factor 2/|ω|.

Action taken:

We thank the referee for the corrections and fixed all minor issues.

Attachment:

main_redline.pdf

---

## Round 3 · Referee Report · Anonymous (Referee 2) · 2025-11-7

Strengths

  1. The paper successfully extends spinDMFT to include quadrupolar interactions, providing an efficient and fully quantum framework for simulating NMR spin dynamics in systems with S>1/2.

  2. The presentation is systematic, supported by clear figures, quantitative analyses, and openly available code, ensuring transparency and accessibility.

  3. The results demonstrate how quadrupolar terms alter relaxation and line shapes, highlighting the limits of classical approximations and establishing a foundation for future realistic NMR modeling.

Weaknesses

  1. The inclusion of a local quadrupolar term, while useful, is a natural and technically straightforward extension of previous spinDMFT work rather than a conceptual breakthrough.

  2. The results are not benchmarked against exact diagonalization, hybrid simulations, or experimental data, leaving the quantitative accuracy of the approach untested.

  3. The analysis is restricted to homogeneous systems at infinite temperature, without exploring finite-temperature effects, multi-species systems, or experimental comparisons that would broaden its impact.

Report

In my view, this manuscript meets the publication standards of SciPost Physics. It presents a timely and well-motivated study that extends spin dynamic mean-field theory to include quadrupolar interactions, thereby enabling efficient quantum simulations of NMR spin dynamics in systems with S>1/2. The work is methodologically rigorous, conceptually clear, and reproducible, with open-source code and transparent documentation of the self-consistency procedure. The results yield physical insight, showing how quadrupolar terms qualitatively alter relaxation and spectral line shapes while exposing the limitations of classical spin dynamics. The comparison between quantum and classical results is elegant and pedagogically valuable. By combining theoretical innovation, computational efficiency, and direct physical relevance, this work represents a advancement in our understanding of quantum spin dynamics and thus meets the originality, clarity, and significance criteria for publication in SciPost Physics.

Requested changes

  1. Include a short comparison—numerical or qualitative—with known results or experimental trends.

  2. Add one or two sentences on possible future applications or extensions.

Recommendation

Publish (easily meets expectations and criteria for this Journal; among top 50%)

  • validity: top
  • significance: high
  • originality: high
  • clarity: top
  • formatting: excellent
  • grammar: excellent

Author:  Timo Gräßer  on 2025-11-27  [id 6083]

(in reply to Report 1 on 2025-11-07)

General reply:

We thank the Referee for thoroughly reading our manuscript and for the constructive comments. We are very glad about the positive feedback and recommendation for publication in SciPost Physics. We understand the Referee's concern regarding the lack of a qualitative comparison of our data to experiment. This was also the main point of criticism of the first Referee reading our manuscript. We decided to include a corresponding section.

1. requested change:

Include a short comparison-numerical or qualitative-with known results or experimental trends.

Reply:

We decided for a direct comparison with experimental data for monocrystalline aluminium nitride (AlN). This system is a bit more complicated than the basic model we considered before, since it contains different nuclear spin species. However, the extension of the self-consistency problem is relatively simple and the numerical implementation straightforward. Moreover, the resulting data agree excellently with the experimental ones. We are convinced that this comparison strongly validates our method and demonstrates its versatility.

Action taken:

We added a new section 4 containing an extension of the model and comparison to experimental data for AlN.

2. requested change:

Add one or two sentences on possible future applications or extensions.

Reply:

The results for AlN support spinDMFT as a prediction tool for determining dipolar broadening of quadrupolar lines.

Action taken:

We added a sentence about this in the abstract and conclusion.

---

## Round 3 · Referee Report · Anonymous (Referee 1) · 2025-12-1

Report

In the revised version of the manuscript (version 4, arXiv:2507.17720v4) the authors answered all questions of the second referee and myself.
In particular, the main criticism of a lack of comparison to other numerical techniques or experiments has been addressed by adding a section on the comparison to experimental FID spectra measured for an AlN single crystal. Whereas the simulated spectra of the ${}^{14}$N spins are in excellent agreement with experiment, some features in the FID spectra of the ${}^{27}$Al spins are less well reproduced. Possible reasons for these deviations are given and discussed. Overall, the added section about the application of spin-DMFT to a real material validates the approach and shows its versatility.
I recommend the manuscript in its current form (v4) for publication in SciPost Physics.

Requested changes

The authors might want to add a sentence in the caption of Figures 7 and 8 introducing the notation $\delta$ for the (chemical) shift.

Recommendation

Publish (easily meets expectations and criteria for this Journal; among top 50%)

---

## Round 3 · Author Response

See reply

---

## Round 5 · Author Response

We would like to thank the editors for organizing the review process and the referees for the thorough reading of our manuscript and the constructive comments. Both referees suggested the inclusion of a comparison to experimental data, at least on a qualitative level. We agree and included a corresponding section in this resubmission.

---

## Round 5 · List of Changes

Major changes:
1.) We added a new section (4) containing an extension of the model and comparison to experimental data for AlN.

Minor changes:
2.) We included a sentence about the experimental comparison in the abstract and a paragraph in the conclusion
3.) We removed one hbar in Eq. (2) to align with the standard NMR convention
4.) We changed the definition of Omega in Eq. (4) to align with the convention of the experiment we are comparing our results to
5.) In the paragraph before Eq. (18), we argue that for an included quadrupolar interaction, the dynamics becomes more local making the autocorrelation a good approximation of the FID
6.) We improved some formulations and removed several typos

---

## Editorial Decision

in_voting